# Differences in Immunogenicity of Three Different Homo- and Heterologous Vaccination Regimens against SARS-CoV-2

**DOI:** 10.3390/vaccines10050649

**Published:** 2022-04-20

**Authors:** Robert Daniel Heinrich Markewitz, David Juhl, Daniela Pauli, Siegfried Görg, Ralf Junker, Jan Rupp, Sarah Engel, Katja Steinhagen, Victor Herbst, Dorinja Zapf, Christina Krüger, Christian Brockmann, Frank Leypoldt, Justina Dargvainiene, Benjamin Schomburg, Shahpour Reza Sharifzadeh, Lukas Salek Nejad, Klaus-Peter Wandinger, Malte Ziemann

**Affiliations:** 1Institute of Clinical Chemistry, University Hospital of Schleswig-Holstein, Arnold-Heller-Straße 3, 24105 Kiel, Germany; daniela.pauli@uksh.de (D.P.); ralf.junker@uksh.de (R.J.); frank.leypoldt@uksh.de (F.L.); justina.dargvainiene@uksh.de (J.D.); benjamin.schomburg@uksh.de (B.S.); sharifzadeh@uksh.de (S.R.S.); lukas.saleknejad@uksh.de (L.S.N.); klaus-peter.wandinger@uksh.de (K.-P.W.); 2Institute for Transfusion Medicine, University Hospital of Schleswig-Holstein, 23538 Lübeck, Germany; david.juhl@uksh.de (D.J.); siegfried.goerg@uksh.de (S.G.); christian.brockmann@uksh.de (C.B.); malte.ziemann@uksh.de (M.Z.); 3Department of Infectious Diseases and Microbiology, University of Lübeck, 23538 Lübeck, Germany; jan.rupp@uksh.de; 4Department of Anesthesiology and Intensive Care, University Hospital of Schleswig-Holstein Campus Lübeck, 23562 Lübeck, Germany; sarah.engel@uksh.de; 5Institute for Experimental Immunology, EUROIMMUN AG, 23560 Lübeck, Germany; k.steinhagen@euroimmun.de (K.S.); v.herbst@euroimmun.de (V.H.); d.zapf@euroimmun.de (D.Z.); ch.krueger@euroimmun.de (C.K.)

**Keywords:** SARS-CoV-2, vaccination, immunogenicity, serology

## Abstract

*Background*: Due to findings on adverse reactions and clinical efficacy of different vaccinations against SARS-CoV-2, the administration of vaccination regimens containing both adenoviral vector vaccines and mRNA-based vaccines has become common. Data are still needed on the direct comparison of immunogenicity for these different regimens. *Methods*: We compared markers for immunogenicity (anti-S1 IgG/IgA, neutralizing antibodies, and T-cell response) with three different vaccination regimens (homologous ChAdOx1 nCoV-19 (n = 103), or mixture of ChAdOx1 nCoV-19 with mRNA-1273 (n = 116) or BNT162b2 (n = 105)) at two time points: the day of the second vaccination as a baseline and 14 days later. *Results*: All examined vaccination regimens elicited measurable immune responses that were significantly enhanced after the second dose. Homologous ChAdOx1 nCoV-19 was markedly inferior in immunogenicity to all other examined regimens after administration of the second dose. Between the heterologous regimens, mRNA-1273 as second dose induced greater antibody responses than BNT162b2, with no difference found for neutralizing antibodies and T-cell response. *Discussion*: While these findings allow no prediction about clinical protection, from an immunological point of view, vaccination against SARS-CoV-2 with an mRNA-based vaccine at one or both time points appears preferable to homologous vaccination with ChAdOx1 nCoV-19. Whether or not the demonstrated differences between the heterologous regimens are of clinical significance will be subject to further research.

## 1. Introduction

Since its inception in late 2019 [1], various measures have been employed in the attempt to contain the pandemic of Coronavirus disease 2019 (COVID-19), caused by the severe acute respiratory syndrome coronavirus 2 (SARS-CoV-2). The most promising of these measures certainly is the vaccination against SARS-CoV-2. Since the end of 2020, several vaccines, using different modes of action on the molecular level, have been approved and administered in most parts of the world, among them the adenoviral vector vaccine ChAdOx1 nCoV-19 (ChAdOx, AstraZeneca, Cambridge, UK) [2], and the two mRNA vaccines BNT162b2 (BNT, BioNTech/Pfizer, Mainz/New York, NY, Germany/USA) [3] and mRNA-1273 (Mod, Moderna, Cambridge, MA, USA) [4]. As more data concerning efficacy and reactogenicity of these vaccines emerge [5,6,7,8], recommendations regarding the administration of these vaccines change. As of July 2021, the German permanent commission on vaccination (Ständige Impfkommission) recommends for all vaccinees, who have received a first dose of the vaccination with ChAdOx, to receive a second dose of either BNT or Mod (heterologous vaccination). However, recipients can also make the informed decision to receive a second dose of ChAdOx (homologous vaccination). Generally, mainly due to reports of vaccine-induced immune thrombotic thrombocytopenia (VITT) in predominantly younger (<60 years) female recipients of ChAdOx [8], in Germany, the use of ChAdOx is recommended only for recipients ≥ 60 years. To date, several studies have shown that heterologous vaccinations are non-inferior to homologous vaccinations with ChAdOx. On the contrary, data have shown that some immune responses to heterologous vaccinations may be even greater than those to homologous vaccinations [9,10,11,12].

During a previous study, examining homologous mRNA-based vaccination regimens, we saw that vaccination with mRNA-1273 was associated with stronger immune responses, compared with BNT162b2, while there was an inverse correlation between age and the examined immune responses [13]. In another study, we examined the kinetics of the B-cell response to a second dose of three different vaccines against SARS-CoV-2 in a cohort of participants who had previously received a first dose of ChAdOx. We found that within the first seven days after the second vaccination, a second dose of Mod or BNT elicits earlier and stronger B-cell responses than a second dose of ChAdOx, while a trend showed stronger reactions after a second vaccination with Mod compared to BNT [14].

This study aims to expand the current knowledge on heterologous vaccinations against SARS-CoV-2 by comparing both regimens of heterologous vaccination currently practiced in Germany with the homologous vaccination with ChAdOx. To this end, we examined several markers for the immune response to the different vaccination regimens in different cohorts of vaccinees. Drawing on available data, we hypothesized that:In continuation of the trend already seen within the first seven days after the second vaccination, heterologous vaccinations induce an immune response comparable or superior in size to that induced by homologous vaccination with ChAdOx.Among heterologous vaccinations, a second dose of Mod induces a greater immune response than a second dose of BNT (in analogy to the corresponding homologous vaccination regimens [13] and the trend we have seen within the first seven days [14]).Age might be a general influencing factor, with older age possibly being associated with weaker immune responses.

## 2. Methods

### 2.1. Study Population

For this observational (i.e., non-randomized) study, participants (all in all 360) were recruited from individuals receiving their vaccination against SARS-CoV-2 at the University Hospital Schleswig-Holstein (Lübeck, Germany). The study population represents a sample of convenience and no statistical sample size considerations were performed beforehand. Three cohorts were selected: the first cohort consisted of vaccinees who received two doses of ChAdOx (ChAdOx/ChAdOx); the second cohort of those who received a first dose of ChAdOx and a second one of Mod (ChAdOx/Mod); and the third cohort of those who received a first dose of ChAdOx and a second one of BNT (ChAdOx/BNT). More information on the vaccination rollout and the recruitment of participants can be found in the supplement. Figure 1 represents a visualization of the process for vaccination and recruitment. As was expected due to the abovementioned recommendation by Germany health authorities, the age distribution among participants receiving ChAdOx/ChAdOx included more persons 60 years of age or older (see Figure 2).

The interval between both doses was 12 weeks for all three cohorts. The participants were asked to disclose any immunosuppressive medication; five participants confirmed this (the respective substances being: Budesonid (per inhalationem), Rituximab, Infliximab, Ciclosporin + Dupilumab, and Methotrexate. Due to German recommendations concerning the vaccination regimen in different age groups, there was a slightly unequal age distribution among the three cohorts for participants 60 years and older. No information on adverse events was solicited from the participants and none was received unsolicited. In order to ensure an equal distribution among age groups and sexes among the three cohorts, only participants younger than 60 years of age (n = 324) were included in the statistical analysis of differences between the cohorts. The analysis of the data of the 36 participants 60 years of age and older can nonetheless be found in the Appendix A.

Prior to participation, all participants gave written informed consent to all procedures they underwent. The study was approved by the University of Kiel institutional review board (AZ: D642/20) and performed in accordance with the declaration of Helsinki [15].

### 2.2. Sample Characteristics

Blood samples were collected from the participants at two different time points: immediately prior to receiving the second dose of the vaccination against SARS-CoV-2 (t_1_; in order to assess the serological status after having received one dose of ChAdOx) and two weeks after this (t_2_). At each of these two time points, a serum and a lithium–heparinate blood sample were collected from each participant. The samples were subsequently pseudonymized before tests were performed.

### 2.3. Anti-SARS-CoV-2 Antibodies

Antibodies of the classes IgA and IgG against the S1 subunit of the Spike protein of SARS-CoV-2 (anti-S1 IgG) as well as IgG against the nucleocapsid (NCP) antigen were measured from the serum samples using the Anti-SARS-CoV-2-ELISA IgA and Anti-SARS-CoV-2-QuantiVac-ELISA (IgG) test kits by EUROIMMUN (Lübeck, Germany) according to the manufacturers’ instructions. While the IgA test kit yields a semiquantitative test result reported as the optical density (OD) of the sample compared to that of a control, the IgG test kit yields a quantitative result reported in binding antibody units per milliliter (BAU/mL), ensuring international comparability of the results. For anti-S1 IgA and anti-NCP IgG, values of >1.1 OD were considered as reactive, values of <0.8 OD as negative and all values in between as borderline. For anti-S1 IgG, values of >35.2 BAU/mL were considered as reactive, values of <25.6 BAU/mL as negative and all values in between as borderline.

### 2.4. Surrogate Neutralization Assay

The capacity of the anti-S1 IgG to potentially neutralize SARS-CoV-2 was tested via a surrogate neutralization assay (NeutraLISA, EUROIMMUN, Lübeck, Germany). The surrogate neutralization assay (NeutraLISA, EUROIMMUN, Lübeck, Germany) is a competitive ELISA in which soluble biotinylated ACE2, contained in a buffer, competes with anti-S1 IgG in the sample for the binding at the recombinant S1 subunit of the Spike protein of SARS-CoV-2 that is attached to a solid phase. The amount of ACE2 that is detected after a washing step via an enzymatic reaction, employing streptavidin-coupled peroxidase, is inversely proportional to the concentration of neutralizing antibodies. The signal detected in samples is compared to the mean signal of a duplicate measurement of the buffer alone (blank), for which the highest possible signal is expected. The result is reported as inhibition [%], which is calculated via the following formula: $\mathrm{inhibition}\;\left[\%\right]=100\%-\frac{\mathrm{extinction}\;\mathrm{of}\;\mathrm{the}\;\mathrm{sample}\times 100\%}{\mathrm{mean}\;\mathrm{extinction}\;\mathrm{of}\;\mathrm{the}\;\mathrm{blank}}$. Of note, by design, this assay has an upper limit of quantification at 100%. Differences in concentrations of neutralizing antibodies beyond this point are not quantifiable via this assay. Values < 20% are considered non-reactive, values ≥ 35 are considered reactive and values between 20 and 35% are considered to be borderline. According to the manufacturer, there is a concordance of 98.6% between this method and the examination of neutralizing antibodies via plaque reduction neutralization test (PRNT50) [16].

### 2.5. Interferon-γ Release Assay

An Interferon-γ-release assay (IGRA) was performed as a marker for the recipients’ T-cell response to the vaccination against SARS-CoV-2. This was carried out using assays by EUROIMMUN (SARS-CoV-2 IGRA stimulation tube set, Lübeck, Germany). For the Interferon-γ-release assay (IGRA), lithium–heparinate anticoagulated whole blood, collected from the participants, was divided into three aliquots undergoing different stimulation protocols for 20–24 h: one aliquot being stimulated with SARS-CoV-2-specific antigens (a mixture of antigens based on the n-terminal domain of the spike protein including the RBD region); another being stimulated with a lectin-based mitogen; and a third aliquot (blank) undergoing no stimulation at all (as a baseline control). Subsequent to these stimulation steps, all aliquots were centrifuged and interferon-γ was measured in the supernatant plasma using an ELISA by EUROIMMUN (Lübeck, Germany). The IGRA yields a quantitative level of measured Interferon-γ, reported in mIU/mL. Interferon-γ values of >200 mIU/mL were considered as reactive, 100–200 mIU/mL are considered borderline and <100 mIU/mL as negative. Apart from the IGRA, we did not perform further analyses of the T-cell response quantifying the contribution for different subsets of T-cells (such as CD4^+^ of CD8^+^ T-cells) to the detected Interferon-γ release. However, previous studies have shown that stimulation with SARS-CoV-2 peptide pools elicits a broad T-cell responses encompassing both CD4^+^ and CD8^+^ T-cells (among others) [17,18,19].

### 2.6. Statistical Analysis

As a first step, normality of the data was assessed via the Shapiro–Wilk test. As all examined variables were non-normally distributed, non-parametric testing was applied for further statistical analysis. For the analysis of differences in continuous variables between multiple groups, the Kruskal–Wallis test was calculated. If the influence of more than one categorical independent variable was examined, *p*-values were adjusted for multiple comparisons using the method described by Benjamini and Yekutieli [20]. If this exploratory analysis showed significant main or interaction effects for the examined factors, post-hoc testing via Dunn’s test, a pairwise Mann–Whitney test with Benjamini–Yekutieli correction for multiple comparisons, was applied. For differences between two groups, the Mann–Whitney *U* test was used. For the analysis of the distribution among a categorically scaled variable between different groups, Pearson’s chi-squared test was used. For the analysis of the association between two continuous variables, correlations using Spearman’s rho were calculated. This test was chosen due to its ability to identify non-linear correlations, as were detected during the previous study [13]. Statistical significance was assumed for *p*-values < 0.05. Wherever average values with a measure of dispersion are reported, these represent the medians and the median absolute deviation (MAD), unless otherwise stated. All statistical analyses were performed using the open-source software for statistical computing and graphics R (version 4.1.0; R Core Team; Vienna, Austria) with the integrated development environment RStudio (Version 1.4.1717; R Core Team; Vienna, Austria) [21].

## 3. Results

### 3.1. Study Population

Overall, 324 participants were included in the statistical analysis, of whom 178 (54.9%) were female. Of these 324, 103 participants (31.8%) received ChAdOx/ChAdOx, 116 (35.8%) received ChAdOx/Mod, and 105 (32.4%) received ChAdOx/BNT. The participants’ mean age was 39.3 (sd: ±11.7; range: 18–59; see Figure 2 for a histogram of participants’ ages for each different vaccination regimen). For details on the demographical data of the participants, see Appendix A. The t_1_ data from all 324 participants is available for all biomarkers. Due to loss to follow-up, at t_2_, data are available for 304 participants (ChAdOx/ChAdOx: 95; ChAdOx/Mod: 109; ChAdOx/BNT: 100). All participants showed sufficient cellular responses to the control mitogens in the IGRA. A Kruskal–Wallis test with the factor vaccination regimen revealed no statistically significant effect on participants’ ages (*p* = 0.07), as did a Mann–Whitney *U* test with the factor sex (*p* = 0.572). Pearson’s chi-squared test revealed that there was no statistically significant difference in gender distribution between the different vaccination regimens (chi-squared = 0.30179, df = 2, *p*-value = 0.8599).

### 3.2. Effect of the Second Dose

The comparison among all examined markers across all vaccination regimens of the current study revealed a significant increase in these markers from t_1_ to t_2_ (*p* < 0.001–0.0001; see Figure 3A–L). For average results of all markers at both time points and for all subgroups, please see Table 1.

### 3.3. Influence of Sex on the Immune Response

Mann–Whitney *U* tests revealed that there were differences between the two sexes in both IgG and IgA response, whereas for anti-S1 IgG, there were slightly stronger responses for female participants at t_2_ (*p* = 0.0442), for anti-S1 IgA, there were slightly stronger responses for male participants, both at t_1_ (*p* = 0.0001) and at t_2_ (*p* = 0.0331). For both neutralizing antibodies and for the IGRA, no statistically significant difference could be found between the two sexes (see Figure 4).

### 3.4. Influence of the Vaccination Regimen on the Immune Response

Kruskal–Wallis tests with the factor vaccination regimen revealed a statistically significant effect on the levels of all examined markers at t_2_ (*p* < 0.0001 in all cases). Post-hoc testing via Dunn’s test revealed that for anti-S1 IgG and IgA, as well as for neutralizing antibodies, comparisons of all three vaccination regimens in the current study with one another yield statistically significant differences (*p* < 0.0001 for all comparisons between ChAdOx/ChAdOx and both heterologous regimens; the comparison between ChAdOx/Mod and ChAdOx/BNT yielding somewhat smaller levels of significance at t_2_: anti-S1 IgG: *p* = 0.001; anti-S1 IgA: *p* = 0.006; neutralizing antibodies: *p* = 0.041). For all of these three markers, levels were highest after ChAdOx/Mod, followed by ChAdOx/BNT and ChAdOx/ChAdOx. For the IGRA, comparisons between ChAdOx/ChAdOx and both heterologous regimens also revealed highly significant differences (*p* < 0.0001) with stronger results for the heterologous regimens, but no statistically significant difference between both heterologous regimens (*p* = 0.812; see Figure 5).

At t_1_, Kruskal–Wallis tests showed a statistically significant effect for the vaccination regimen (i.e., for the type of second vaccine, which the participants had not yet received) only for anti-S1 IgA (*p* = 0.033) and neutralizing antibodies (*p* = 0.018). Post-hoc testing via Dunn’s test revealed no statistically significant difference between any of the regimens for anti-S1 IgA, while it revealed a small but statistically significant difference in neutralizing antibodies between participants who would later receive Mod or BNT, with somewhat stronger reaction for the former (19 ± 16.3% vs. 13 ± 11.9%; *p* = 0.044).

### 3.5. Influence of Age on the Immune Response

There was no correlation in meaningful effect size between any of the examined markers and the participants’ age, neither after the first, nor after the second dose of the vaccination (absolute rho ≤ 0.2 for all calculated correlations). In analogy, when participants are grouped together by life decade, there are only a few comparisons that reveal statistically significant differences in the results of the different markers for each life decade (specifically: the comparisons between results for participants in the 2nd life decade with those for the 3rd, 4th, and 5th life decade reveal higher levels of anti-S1 IgG for the 2nd life decade at t_1_ (*p*-values: 0.007, 0.014, and 0.007, respectively), while the same is true for the comparison between the 3rd and the 5th life decade at t_2_, revealing higher levels of neutralizing antibodies for the former group (*p* = 0.042).

### 3.6. Influence of Prior Infection on the Immune Response

At t_1_, there were nine participants who tested positive for anti-NCP IgG, suggesting that they had been exposed to SARS-CoV-2 at some earlier point in time. All of these nine participants tested positive again at t_2_; two received ChAdOx/ChAdOx, three received ChAdOx/BNT, and four received ChAdOx/Mod. There were no participants who tested positive at t_2_ after testing negative at t_1_, suggesting that none of the participants had contracted SARS-CoV-2 in the meantime.

Despite the very low number of previously exposed individuals, Mann–Whitney *U* tests showed that these had significantly higher levels of anti-S1 IgG (*p* = 0.0004), anti-S1 IgA (*p* = 0.0002), inhibition (via surrogate neutralization assay; *p* = 0.002), and IFN-γ (via IGRA; *p* = 0.003) at t_1_ than participants who tested anti-NCP negative (although with great interindividual fluctuations). At t_2_, none of these comparisons revealed a statistically significant difference. When comparing the results for previously infected participants after the first dose with those of unexposed individuals after the second dose (in order to see if one dose of ChAdOx and a previous or following COVID-infection result in a similar immune response as two doses of vaccines for unexposed individuals), we found that both anti-S1 IgG and neutralizing antibodies were significantly lower (*p* = 0.008 and *p* = 0.004, respectively) for previously infected participants at t_1_ than for unexposed participants at t_2_, with no significant difference for anti-S1 IgA and IFN-γ. When the same comparison was made with only those unexposed participants who received ChAdOx as a second dose, no significant differences could be found, while the comparison with those uninfected participants who had received either BNT or Mod showed significantly stronger reactions for the unexposed individuals for all comparisons.

### 3.7. Borderline or Non-Reactive Samples

For all examined markers as well as at both time points, we observed samples that returned only borderline or negative results. A detailed analysis of these samples can be found in the Appendix A. In short, participants who tested borderline or negative at t_2_ already had on average lower levels of the same markers at t_1_ when compared to participants who donated reactive samples at t_2_. This difference was apparently not mediated by age.

## 4. Discussion

Our results show that all examined vaccination regimens are immunogenic and that the second dose of the vaccine acts as a booster with significant increases in the levels of all examined markers after the second vaccination. However, the results also reveal significant differences between the different vaccination regimens: no matter which marker is examined, the heterologous vaccination regimens with an mRNA-based vaccine as second dose induce significantly greater reactions than ChAdOx/ChAdOx. The median levels of anti-S1 IgG and Interferon-γ (via IGRA) are on average tenfold higher for the heterologous vaccination regimens than for ChAdOx/ChAdOx, while levels of antibody-induced inhibition are considerably lower and more dispersed for the latter. In addition, a small, but non-negligible number of participants, especially receiving ChAdOx/ChAdOx, did not develop levels of the examined markers considered to be reactive. This is a clear continuation in the results we saw within the first 7 days after the second vaccination [14], with the exception that after 14 days, we were able to detect significant increases in B- and T-cell responses after a second vaccination with ChAdOx, while we were not able to detect significant increases for most markers within the first 7 days.

It has been shown in the course of several studies to date that heterologous vaccination regimens against SARS-CoV-2 are safe and immunogenic [9,10,11,12,22,23,24,25]. All available studies support our finding that heterologous vaccination regimens (ChAdOx/BNT or ChAdOx/Mod) elicit a stronger immune response than ChAdOx/ChAdOx. Where this was examined [9,10,11,12,22], heterologous regimens (mostly ChAdOx/BNT) were also found to elicit immune responses comparable to those elicited by BNT/BNT. Interestingly several of these other studies demonstrated at least partially stronger immune responses to heterologous regimens compared to homologous ones [9,10,11,12,26]. A weakness of all previous studies is that none of them examined the regimens ChAdOx/Mod or Mod/Mod separately. Those who included participants receiving Mod rather pooled the results with those of the corresponding BNT regimen, which might skew the resulting data, as we could show significant differences between ChAdOx/Mod and ChAdOx/BNT, with the stronger responses, at least for anti-S1 IgA and IgG, for the former. This difference between ChAdOx/Mod and ChAdOx/BNT might be a result of a higher mRNA concentration in the former (100 µg vs. 30 µg, according to the manufacturers).

When discussing these data, it is important to bear in mind that they only represent measures for vaccinees’ immune responses to the different vaccination regimens and that it is difficult at this point in time to infer conclusions about a recipient’s protection from SARS-CoV-2 from these data. While it is convenient to assume that higher levels, especially of anti-S1 IgG and antibody-induced inhibition, confer a higher degree of clinical protection from SARS-CoV-2, evidence to support this claim is still scarce. As of yet, levels of neutralizing antibodies likely permit the best prediction of protective immunity from SARS-CoV-2 [27,28], while there is also evidence that higher levels of anti-S1 IgG and T-cell responses are associated with protection from SARS-CoV-2 [29]. Furthermore, several studies have shown that anti-S1 IgG levels correlate well with the antibodies’ ability to neutralize the virus [30]. As there is evidence that antibodies against SARS-CoV-2 tend to wane over time, both after vaccination [31] and infection with SARS-CoV-2 [32,33], higher initial levels of anti-SARS-CoV-2 antibodies in all likelihood provide protection over a longer period of time than do lower levels. Compounding this, antibody levels induced by ChAdOx wane at a higher pace than those induced by BNT [31]. The fact that ChAdOx/ChAdOx induces lower levels of anti-SARS-CoV-2 antibodies than do mRNA-based vaccines, and that these levels furthermore appear to wane faster might explain why clinical efficacy is higher for Mod/Mod and BNT/BNT than for ChAdOx/ChAdOx [2,3,4]. All this suggests that vaccination regimens, either homologous or heterologous, employing mRNA-based vaccines against SARS-CoV-2, are more effective and likely provide better protection from SARS-CoV-2 than ChAdOx/ChAdOx.

As mentioned, we could show stronger immune responses at t_2_ for ChAdOx/Mod, compared to ChAdOx/BNT for all examined markers except for IFN-γ, probably as a consequence of the higher dose of mRNA in Mod. Whether or not this translates to better clinical protection from SARS-CoV-2 remains questionable, as the differences are not as pronounced. The same is true for the differences we previously found between the homologous mRNA-based regimens (Mod/Mod and BNT/BNT) and their heterologous counterparts [13]. In this study, there is also a possible bias towards Mod, as we could show slightly higher levels of neutralizing antibodies at t_1_ already in participants who would later receive Mod, compared to those who would receive BNT.

Interestingly, contrary to findings regarding homologous mRNA-based regimens [34], including our own [13], we could find no relevant influence of age on any part of the immune response in the current study. It is possible that age is a more important influencing factor for homologous vaccinations than for heterologous vaccinations, but it is also possible that the influences of age on the immune response to vaccination against SARS-CoV-2 decreases over time, which is supported by the fact that the negative correlation between the examined markers and age was stronger two weeks after the first vaccination than after the second vaccination in the previous study [13]. However, it is also possible that we missed the potential effects of very old age, as the oldest participant recruited was 70 years old and a previous study saw a marked drop in antibody levels for vaccines older than 80 years old [34].

The analysis of non-reactive or borderline results for any marker shows that, for most participants, two effects must interact to lead to these results: certain persons (independent of their age) appear to show generally weaker responses to the vaccination (which is apparent from them already showing weaker responses after the first vaccination than the rest of the cohort). However, for most, it is unclear what exactly causes these weaker responses, apart from a small number of participants who reported receiving immunosuppressive medication. In some of these persons, a second dose of ChAdOx (as opposed to Mod or BNT) might not be immunogenic enough to elicit significant immune responses. The fact that there was no difference in immune responses between the ChAdOx/ChAdOx sub-cohort and the other two sub-cohorts after the first dose (except for the small difference in neutralizing antibodies between ChAdOx/ChAdOx and ChAdOx/Mod) suggests that the general distribution among potential weak responders was not skewed towards ChAdOx/ChAdOx, but that they were present in recipients of the other two regimens as well.

We saw those participants likely to have been exposed to SARS-CoV-2 as indicated by anti-NCP IgG positivity developed higher levels of most examined markers for immunity already at t_1_. In the previous study, we observed a similar effect [13] and it is generally known that persons who have been exposed to SARS-CoV-2 develop much higher titers of anti-SARS-CoV-2 antibodies after the first dose of the vaccination than unexposed individuals [35,36,37,38]. The comparison of results for previously infected participants after the first dose and unexposed participants after the second dose shows that the combination of one dose of ChAdOx and an infection has a similar boostering effect as a second dose of ChAdOx has for unexposed individuals who have already received one dose of ChAdOx. However, we could also observe that heterologous vaccination (ChAdOx/BNT or ChAdOx/Mod) in unexposed individuals leads to stronger responses of all examined markers compared to the combined effect of one dose of ChAdOx and an infection. Therefore, it seems advisable that previously infected individuals who have received one dose of ChAdOx and individuals infected after the first ChAdOx dose should also receive a second dose (preferably BNT or Mod) as a boost in their immune response.

Finally, there is the difference in anti-S1 IgG and IgA levels between the sexes. However, this difference can only be found for all regimens cumulatively with no significant differences for each regimen separately, and it is not uniform for IgG (which is higher in women only at t_2_) and IgA (which is higher in men at both t_1_ and t_2_). There is evidence for other vaccines that women generally tend to develop stronger humoral reactions, especially IgG, in response to vaccinations [39,40], while there is some evidence that men generally have higher levels of serum IgA than women [41]. Therefore, the reported differences, while statistically significant, might rather be a physiological phenomenon than a SARS-CoV-2 specific effect, and are of questionable clinical significance. Of note, we did not find any sex-associated difference in immune responses to Mod/Mod or BNT/BNT in the previous study [13].

Generally, the role of IgA in respiratory infections and especially the role of the IgA responses in the serum after vaccination is not completely understood [42]. While the early IgA response after the first dose of a vaccination against SARS-CoV-2 (as we have seen in the previous study [13]) might represent a recall response of mucosal B-memory cells formed in response to previous pulmonary coronavirus infections, the response to the second dose is more likely to be dominated by naïve B-cells recruited by the first dose [43]. Whether serum anti-S1 IgA can be used as a marker for mucosal immunity against SARS-CoV-2 (which is important as a first line defense) is therefore unclear and warrants further investigation.

While we did not include recipients of homologous mRNA-based vaccination regimens in the current study, comparisons can be made with the results for the previous study [13], always keeping in mind some methodological differences between the studies (such as different inter-dose intervals): while the antibody-responses (anti-S1 IgG and IgA as well as neutralizing antibodies) appear to be within the same order of magnitude for the heterologous regimens of the current study and the homologous mRNA-based regimens of the previous study (compare the respective tables 1), homologous ChAdOx also elicits markedly weaker responses compared to homologous mRNA-based regimens. This supports findings by others [9,12,22]. Furthermore, while the homologous mRNA-bases vaccines of the previous study appear to have some edge over the heterologous regimens of the current study, this, as well as its clinical significance, would have to be examined further in upcoming studies. A comparison of IGRA-results between the two studies is not possible due to methodological differences.

In conclusion, judging from the markers for assumed immunity against SARS-CoV-2 that we examined, vaccination regimens that employ an mRNA-based vaccine against SARS-CoV-2, either as second dose or for both doses, appear to be preferable to ChAdOx/ChAdOx. Our data do not support superiority of heterologous vaccination regimens over homologous mRNA-based regimens. Rather, the comparison with the previous study [14] shows that, if anything, homologous mRNA-based regimens are likely superior to their heterologous counterparts in the induction of anti-S1 IgG and IgA. However, a detailed analysis of different mRNA-based homo- and heterologous regimens is still lacking. Therefore, while heterologous vaccination regimens might be preferable to ChAdOx/ChAdOx, from our data, it is at best uncertain whether they have an advantage over Mod/Mod or BNT/BNT.

Our study has several limitations. First, no markers for clinical efficacy were examined, but only markers for immunogenicity. The degree of clinical protection conferred by the examined immunological markers cannot be judged from our data. Further, we did not perform a neutralization assay sensu strictu but a surrogate neutralization assay with an upper limit of detection. This hampers comparisons between all vaccination regimens employing mRNA-based vaccines as they elicit very strong reactions in this assay. This is also the cause of the phenomenon that the dispersion of the results for this assay was much greater for ChAdOx/ChAdOx at t_2_ than for the two heterologous regimens, since the former regimens induced significantly weaker results and therefore did not reach the upper limit of detection as the latter two regimens did. Moreover, the period of observation was still short with the last follow-up being 14 days after the second vaccination. Thus, no prediction for the endurance of the markers examined can be made by us. Furthermore, it is possible that immune responses to ChAdOx/ChAdOx are more protracted than responses to mRNA-based regimens. Therefore, comparing all vaccination regimens at the same time point might underestimate the immune responses to ChAdOx/ChAdOx. However, given the size of the differences between ChAdOx/ChAdOx and the other regimens, it is likely that they remain statistically significant independent of the time point for the comparison. Another limitation of our study is that we did perform pre-pandemic serum negative controls in the current study. However, in our previous study, we performed a pre-vaccination serological status, using the very same methods as in the current study, establishing that the status of being non-vaccinated and non-previously infected is reliably associated with non-reactive results in all assays used in the current study [13]. Finally, we did not examine anti-viral vector immunity in our study. This phenomenon might have contributed to the weaker responses to ChAdOx/ChAdOx in comparison with the other regimens and must be explored further in upcoming research.

## Figures and Tables

**Figure 1 vaccines-10-00649-f001:**
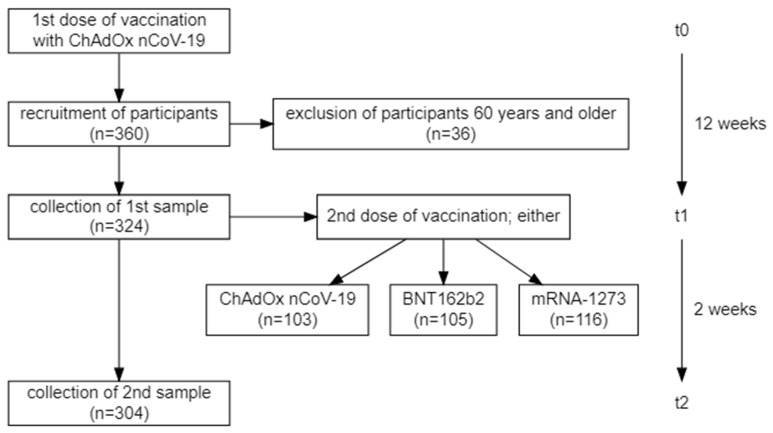
Flow diagram showing the vaccination program and the recruitment of participants for the current study and the respective time intervals in between the different time points.

**Figure 2 vaccines-10-00649-f002:**
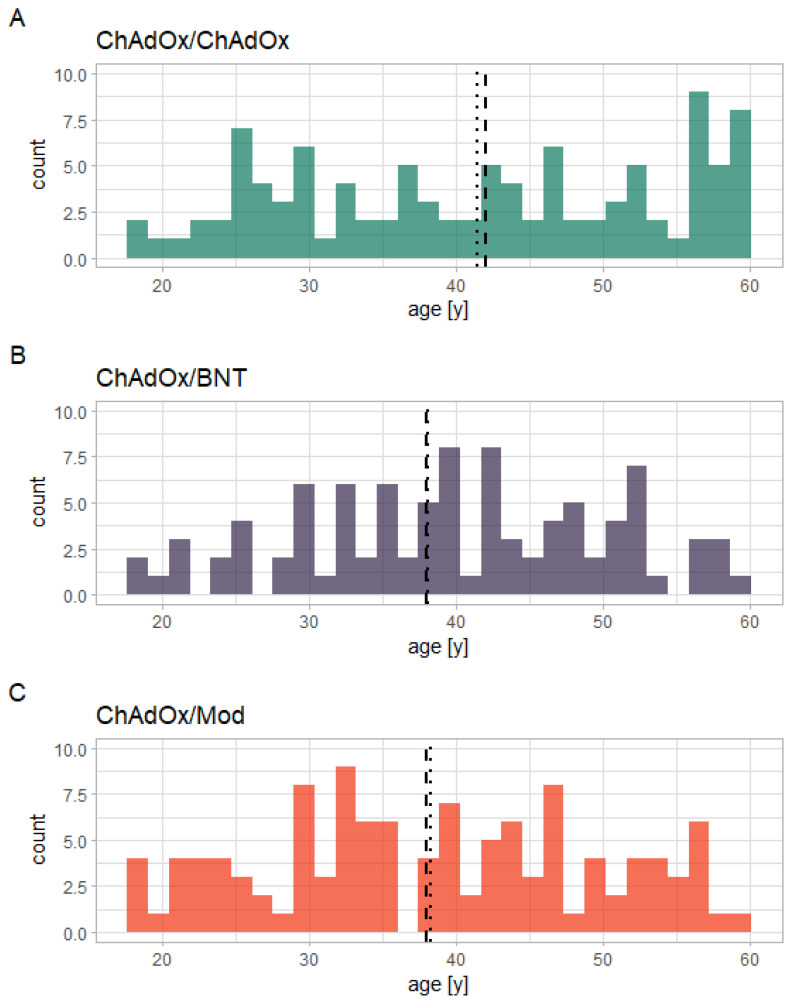
Histogram showing the different age distributions among participants receiving the three different vaccination regimens in the current study: (**A**) ChAdOx/ChAdOx; (**B**) ChAdOx/BNT; (**C**) ChAdOx/Mod). The dashed vertical lines mark the respective median, the dotted lines the mean.

**Figure 3 vaccines-10-00649-f003:**
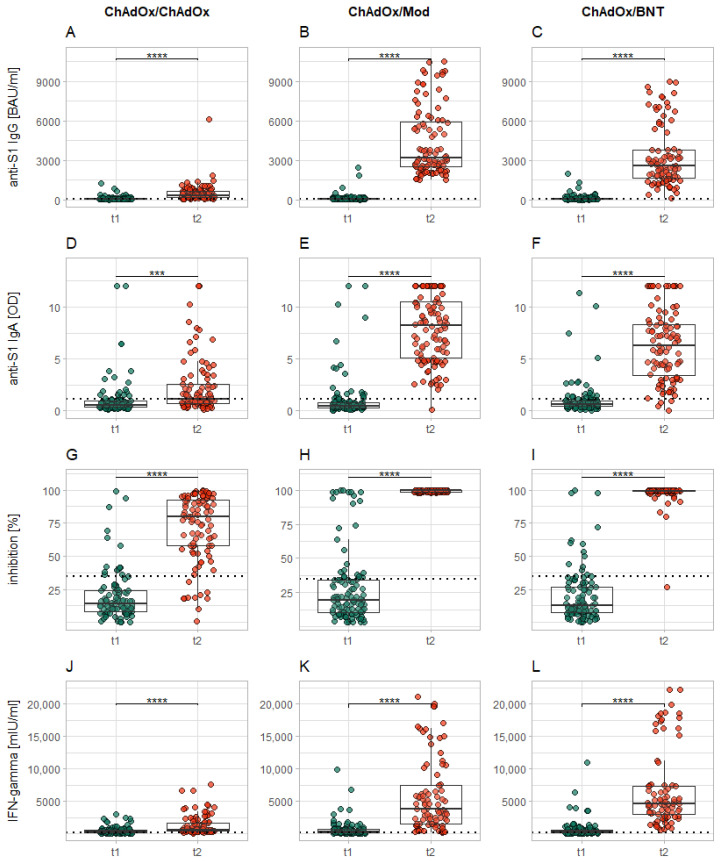
Increase in levels of all examined markers from immediately before the second vaccination (t_1_) to 14 days after it (t_2_). Each column of plots represents the reactions to a specific vaccination regimen (ChAdOx/ChAdOx: (**A**,**D**,**G**,**J**); ChAdOx/Mod: (**B**,**E**,**H**,**K**); ChAdOx/BNT: (**C**,**F**,**I**,**L**)), while each row represents the different examined markers (anti-S1 IgG: (**A**–**C**); anti-S1 IgA: (**D**–**F**); neutralizing antibodies: (**G**–**I)**; IGRA: (**J**–**L**)). The dotted horizontal lines represent the cutoff for reactivity of the different assays. Levels of significance: * = *p* < 0.05; ** = *p* < 0.01; *** = *p* < 0.001; **** = *p* < 0.0001.

**Figure 4 vaccines-10-00649-f004:**
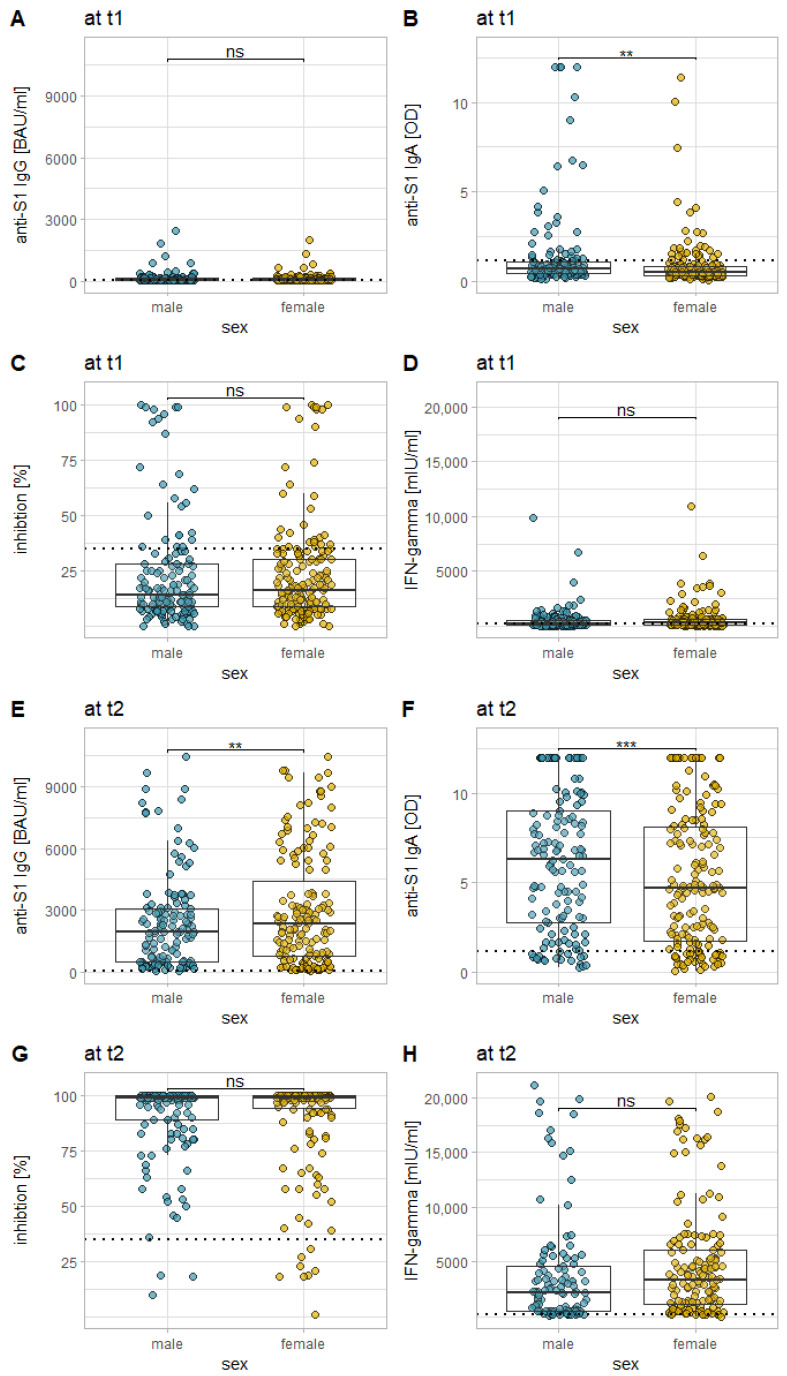
Comparison of the examined immune responses, both 12 weeks after the first vaccination (**A**–**D**), and 14 days after the second vaccination (**E**–**H**) in between the sexes (for all vaccination regimes of the current study cumulatively). Each plot represents a different marker: anti-S1 IgG (**A**,**E**), anti-S1 IgA (**B**,**F**), the inhibition by neutralizing antibodies (**C**,**G**), and T-cell response as measured via IGRA (**D**,**H**). The dotted horizontal lines represent the cutoff for reactivity of the different assays. * = *p* < 0.05; ** = *p* < 0.01; *** = *p* < 0.001; **** = *p* < 0.0001; ns = not statistically significant.

**Figure 5 vaccines-10-00649-f005:**
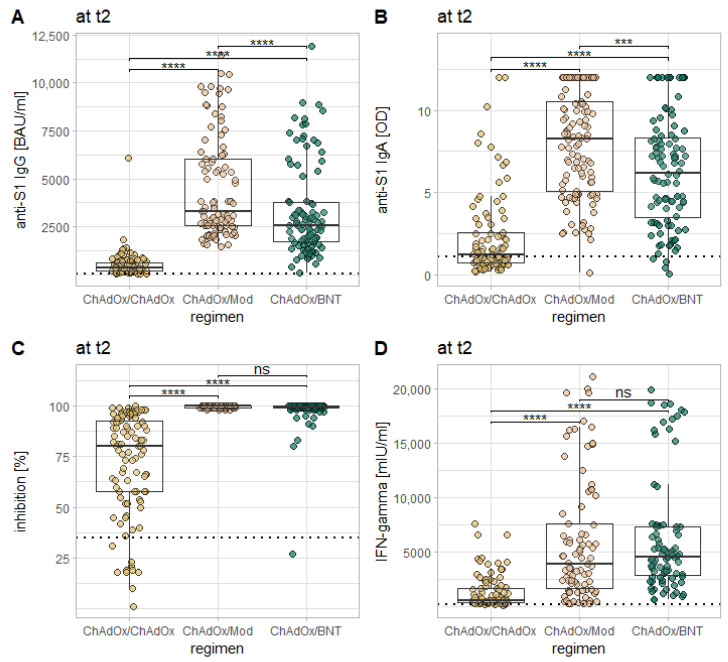
Comparison of the examined immune responses 14 days after the second vaccination in between the three vaccination regimes in the current study. Each plot represents a different marker: anti-S1 IgG (**A**), anti-S1 IgA (**B**), the inhibition by neutralizing antibodies (**C**), and T-cell response as measured via IGRA (**D**). The dotted horizontal lines represent the cutoff for reactivity of the different assays. Levels of significance: * = *p* < 0.05; ** = *p* < 0.01; *** = *p* < 0.001; **** = *p* < 0.0001; ns = not statistically significant.

**Table 1 vaccines-10-00649-t001:** Average values (reported via medians ± median absolute deviation) for all measured markers at all time points, including the respective group sizes, for the whole cohort and the following subgroups: women, men, participants having received ChAdOx/ChAdOx, ChAdOx/Mod, ChAdOx/BNT, and participants who have tested anti-NCP IgG positive at the first time point.

Subgroup	Age (years)	12 Weeks after First Dose				14 Days after Second Dose			
		Anti-S1 IgG (BAU/mL)	Anti-S1 IgA (OD Ratio)	Neutralizing Antibodies (%)	IGRA (mIU/mL)	Anti-S1 IgG (BAU/mL)	Anti-S1 IgA (OD Ratio)	Neutralizing Antibodies (%)	IGRA (mIU/mL)
Whole cohort	39 ± 13.3	63.9 ± 51	0.57 ± 0.36	16 ± 13.3	258 ± 262	2314 ± 2365	5.1 ± 4.9	99 ± 1.5	3289 ± 4028
females	39 ± 13.3	65.2 ± 53	0.5 ± 0.33	16 ± 13.3	280 ± 270	2557 ± 2961	4.7 ± 4.7	99 ± 1.5	3698 ± 4151
males	39.5 ± 15.6	63.0 ± 49.4	0.68 ± 0.44	14 ± 10.4	235 ± 251	2066 ± 2126	6.5 ± 5.0	99 ± 1.5	3025 ± 3747
ChAdOx/ChAdOx	42 ± 17.8	60.5 ± 49.6	0.56 ± 0.34	13 ± 8.9	307 ± 302	358 ± 289	1.3 ± 1	80 ± 23.7	564 ± 465
ChAdOx/Mod	38 ± 12.6	69.1 ± 48.6	0.49 ± 0.31	19 ± 16.3	286 ± 289	3747 ± 2492	8.2 ± 4.1	100 ± 0	5324 ± 6073
ChAdOx/BNT	38 ± 13.3	66.4 ± 58.6	0.66 ± 0.37	13 ± 11.9	235 ± 222	2595 ± 1548	6.3 ± 3.4	99 ± 1.5	4872 ± 3639
Anti-NCP IgG pos.	34 ± 8.9	841 ± 953	3.9 ± 4.6	94 ± 7.4	978 ± 1119	2771 ± 2540	8.3 ± 5.6	99 ± 1.5	7313 ± 8597

## Data Availability

The datasets analyzed during the current study are available from the corresponding author on reasonable request.

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
