# Peer review of "Differences in Immunogenicity of Three Different Homo- and Heterologous Vaccination Regimens against SARS-CoV-2"

_vaccines, 2022, doi:10.3390/vaccines10050649_

Round 1

Reviewer 1 Report

The present contribution is a well written paper with valuable information.

Please comment on:

Table 1 displays values with very high dispersion. Indeed,  SD +/- negative values indicate the calculation included negative values?  Please revise carefully the statistics on the table.

My concern is the lack of interpretation of the high data dispersion in ChAdOx/ChAdOx in detected neutralizing antibodies, IgG antiS1 and IGRA; in contrast when mRNA vaccines were employed.  

I understand no time to include a comparison on mRNA/mRNA vaccination, but it would have provided the point that mixtures are better. 

Author Response

The present contribution is a well written paper with valuable information.

- The authors want to thank the reviewer for their kind assessment of our manuscript, as well as for the opportunity to revise the manuscript following the guidance provided in their suggestions and comments. We hope the revisions and answers provided by us have rendered the manuscript publishable in the eyes of the reviewer. Please find below a detailed point-by point response to each comment made by the reviewer. Should there be any remaining question or concerns, please do not hesitate to tell us so.

Please comment on:

Table 1 displays values with very high dispersion. Indeed,  SD +/- negative values indicate the calculation included negative values?  Please revise carefully the statistics on the table.

- Thank you for your adroit observation. It is true that some values, especially for the IGRA, exhibit quite large measures of dispersion. It is well worth noting, however, that, as a measure of dispersion, we didn’t use the standard deviation (SD) but rather the median absolute deviation (MAD), as mentioned in the statistics section of the Methods. This choice was consciously made as a consequence of the distribution of the data, which did in fact include large dispersions with many outliers, which are not very well represented by the mean and the SD. With the MAD, it is indeed possible to have a dispersion that is greater than the median itself, even if the data only included positive numbers, as was the case for our data. This is the best explanation we can provide for this phenomenon, we hope it is satisfactory for the reviewer.

My concern is the lack of interpretation of the high data dispersion in ChAdOx/ChAdOx in detected neutralizing antibodies, IgG antiS1 and IGRA; in contrast when mRNA vaccines were employed.  

- Again, the reviewer is correct in noting that the dispersion of the data was quite high for most of the results. However, the authors are not sure that they can agree with the statement that the dispersion was greater for ChAdOx/ChAdOx than for the heterologous regimens. In fact, in nearly all cases, the results for ChAdOx/ChAdOx had a smaller (or at least comparable) deviation relative to the median than the heterologous regimens, as can be seen in Table 1. There is one exception to this, which is the examination of neutralizing antibodies at t2. Here, the heterologous regimens have very small dispersions, while ChAdOx/ChAdOx has a comparably large one. This is due, however, to the fact that our surrogate neutralization assay by design had an obvious upper limit of detection at 100 %. Since this assay was designed to assess the serological situation after infection with SARS-CoV-2, in which situation much weaker responses can be observed than after two vaccinations with an mRNA-based vaccine at at least one time point, the surrogate neutralization assay is not ideally suited to asses the serological status after two vaccinations. This is an obvious limitation of this assay and therefore of our study which is duly mentioned in the discussion. However, we have added a sentence to this part of the limitations in which we state that this is the reason why the dispersion in neutralization results is much larger for ChAdOx/ChAdOx than for the other two regimens.

I understand no time to include a comparison on mRNA/mRNA vaccination, but it would have provided the point that mixtures are better.

- The reviewer is absolutely right, a comparison with the corresponding homologous mRNA regimens (BNT/BNT and Mod/Mod) would have been ideal, but was not possible in our cohort for practical reasons. However, it is still possible to compare the numerical data from the current cohort with that of our previous study (Markewitz et al.; “The temporal course of T- and B-cell responses to vaccination with BNT162b2 and mRNA-1273”) in which we have used the same methods at roughly the same temporal intervals. Since this data was already published and since there are some minor methodological differences, we decided against including this data as a reference in the current manuscript (to avoid duplicate publication). We have, however, evaluated this statistically and found that the homologous mRNA-based regimens perform very similar to the heterologous regimens, with even some slight advantages for the homologous regimens in comparison with their respective heterologous counterparts. We do mention this in the discussion. But in the end, it is up to further research to show this comparison in a methodologically sound manner.

Reviewer 2 Report

See attachment

Author Response

The authors want to thank the reviewer for their kind assessment of our manuscript, as well as for the opportunity to revise the manuscript following the guidance provided in their suggestions and comments. We hope the revisions and answers provided by us have rendered the manuscript publishable in the eyes of the reviewer. Please find below a detailed point-by point response to each comment made by the reviewer. Should there be any remaining question or concerns, please do not hesitate to tell us so.

Figure 1 is labelled as Figure 5

  • We thank the reviewer for pointing out this inconsistency, which has been amended in the revised version of the manuscript.

Figure 2: How is it possible that individuals with homologous ChAdOx1 boost, who have very low if undectable anti-S1 IgG (panel A) still have potent neutralization (panel G)? Similar data in Fig 4.

  • The reviewer is right, it may seem as if recipients of homologous ChAdOx1 have next to no anti-S1 IgG after two doses of the vaccine but still they exhibit quite potent levels of neutralization. In order to understand this, one has to examine the data carefully. First of all, it is actually not true that recipients of homologous ChAdOx1 do not exhibit anti-S1 IgG in significant amounts after the second dose of the vaccination. If one looks closely at the data in Table 1, one notices that homologous ChAdOx1 does in fact induce a significant anti-S1 IgG titer of, on average, 358 BAU/ml, which is more than 10x the threshold for reactivity (35.2 BAU/ml; as mentioned in the manuscript) and also more than most patients exhibit after infection with SARS-CoV-2. Therefore, the anti-S1 IgG results after homologous actually cannot really be described as “very low”, except in comparison with the heterologous regimens. But since the y-axes of panels A, B, and C of Figure 2 (as well as panel A of Figure 4) are scaled uniformly, to increase comparability, and since this scale is adapted to present the very high titers seen after the heterologous vaccinations, the titers for homologous ChAdOx1 almost disappear in comparison. The same is not quite true for our neutralization results, however, which is mainly due to the fact that this assay was designed to assess the serological status after infection with SARS-CoV-2, in which situation much lower titers are usually observed than after two vaccinations, at least one of which being an mRNA-based vaccine. This causes the phenomenon that neutralization results for homologous ChAdOx1 are actually quite well represented with the surrogate neutralization assay while the results for the heterologous regimens are of somewhat lesser informative value as they are very uniformly positioned at the upper limit of detection for this assay. This is of course a limitation of this assay and it is being discussed as such. If the reviewer is further interested in the relationship between the anti-S1 IgG assay and the surrogate neutralization assay we used, we recommend our paper, which was recently published, entitled: “Kinetics of the Antibody Response to Boostering With Three Different Vaccines Against SARS-CoV-2” (Markewitz et al., 2022; also mentioned in the current manuscript). Here we examined the kinetics of the antibody responses in very similar cohorts of vaccinees in the first seven days after the second vaccination and we saw that for homologous ChAdOx1, the neutralization assay detects a response earlier and more strongly than anti-S1 IgG, which might be caused by the surrogate neutralization assay being, as a result of its’ design, more sensitive for changes in the antibody response than the anti-S1 IgG assay.

A pre-pandemic serum negative control seems to be lacking.

  • The reviewer is correct, we did not perform a pre-pandemic serum negative control in the current study. However, in the previous study, we performed pre-vaccination controls of the serological status of the participant, establishing that the status of being non-vaccinated an non-previously infected is reliably associated with non-reactive results in all assays used in the current study as well. We have included a paragraph mentioning this limitation at the end of the Discussion.

It would be informative to know whether anti viral vector immunity was developed and how this correlated with the magnitude of the boost.

  • We agree with the reviewer that the examination of anti-vector immunity would have added valuable insight to our data. However, this was not a focus of our study and will have to be addressed in further research. We have added this to the limitations section in the Discussion.

How were the cellular responses to control mitogens?

  • As mentioned in the Methods section, we performed the IGRA according to the manufacturer’s instruction. In this case this entails that samples which do not show a cellular response to the control mitogens (for whatever reason) would have been excluded from further analysis. However, all samples exhibited sufficient cellular responses to the control mitogens. We have added this information to the results section.

I could not see the supplementary figure.

  • We are sorry to read that you couldn’t see supplementary Figure S1. We have made sure that it is included in the folder with the supplementary material this time and want to apologize for this apparent lack in oversight on our part.